# A Simple Strategy for the Simultaneous Determination of Dopamine, Uric Acid, L-Tryptophan and Theophylline Based on a Carbon Nano-Onions Modified Electrode

Rui An, Wenzhu Kuang, Zijian Li, Tiancheng Mu  and Hongxia Luo *

Department of Chemistry, Renmin University of China, Beijing 100872, China; 2020102286@ruc.edu.cn (R.A.); 2021102232@ruc.edu.cn (W.K.); 2022102185@ruc.edu.cn (Z.L.); tcmu@ruc.edu.cn (T.M.)
* Correspondence: luohx@ruc.edu.cn; Tel.: +86-10-62512822

**Abstract:** In this work, carbon nano-onions (CNOs) with particle sizes of 5–10 nm were prepared by the multi-potential step method. High-resolution transmission electron microscopy, infrared spectroscopy and Raman spectroscopy characterize the effective synthesis of CNOs. CNOs/GCEs were prepared by depositing the prepared CNOs onto glassy carbon electrodes (GCEs) by a drop-coating method. Examination of the electrocatalytic activity of the CNOs/GCE sensor by simultaneously detecting dopamine (DA), uric acid (UA), L-tryptophan (Trp) and theophylline (TP) using a differential pulse voltammetry technique. The results showed that the linear ranges of DA, UA, Trp and TP were DA 0.01–38.16 μM, UA 0.06–68.16 μM, Trp 1.00–108.25 μM, and TP 8.16–108.25 μM, and the detection limits (S/N = 3) were 0.0039 μM, 0.0087 μM, 0.18 μM and 0.35 μM, respectively. The $CNO_S$/GCE sensor had good stability and could be used for the detection of actual samples.

**Keywords:** carbon nano-onions; electrochemical sensing; glassy carbon electrode; theophylline; uric acid; L-tryptophan; dopamine

## 1. Introduction

There are a variety of biomolecules in the human body, and the concentration data of specific biomolecules can provide crucial reference values in the initial diagnosis of diseases. Dopamine (DA) in the central nervous system is an excitatory neurotransmitter belonging to the monoamine class and plays a critical role in human physiological functions [1]. The imbalance of dopamine concentration will affect the normal operation of the human central nervous system and hormone system, which will lead to Parkinson's disease, schizophrenia, Tourette's syndrome, Attention Deficit Hyperactivity Syndrome and pituitary tumourigenesis [2,3]. Uric acid (UA) is a protein metabolite found in human urine, blood and other biological fluids and is a human health hazard [4]. Abnormal UA levels in the body can cause hypertension, metabolic syndrome, kidney injury, and other diseases [5]. L-tryptophan (Trp), one of the 20 amino acids essential for protein biosynthesis, has major effects on human metabolism, including nitrogen balance, muscle mass, and weight maintenance [6]. Theophylline (TP) is a natural alkaloid that can produce various physiological effects, such as relaxing bronchial muscles, secreting gastric acid, and stimulating the central nervous system. It is effective in treating bronchial asthma in clinics due to its ability to relax smooth muscle [7]. DA, UA, Trp and TP co-exist in human serum. Since the detection of these four biomolecules is of great importance not only in diagnostics and pathology research but also in biomedicine and neurochemistry, their efficient simultaneous identification and characterization are becoming more and more popular and critical in many different fields.

Due to the importance of these four biomolecules, there are various materials used for the study of the electrochemical analysis of these four biomolecules, but the main focus is on the differentiation of two or more of DA, UA, Trp and TP at different electrodes.

One of the most widely used methods to solve this problem is the use of chemically modified electrodes. These include metal nanoparticle-modified electrodes [8–10], polymer-modified electrodes [11,12], carbon nanotubes and their complex-modified electrodes [13] and graphene and its complex-modified electrodes [14,15].

Zhang et al. [16] produced nitrogen-doped graphene by thermal depolymerization (p-phenylenediamine) complexes, and nitrogen-doped graphene showed excellent sensing performance for the simultaneous detection of DA, AA and UA. Yang et al. used ferrocene derivatives functionalized with AuNP/CNOs nanocomposites and graphene to differentiate ascorbic acid (AA), DA, UA and acetaminophen [17]. Chen et al. modified poly(β-cyclodextrin) and CNOs on GCE to effectively detect DA, UA and Trp simultaneously [18]. Sun et al. reported a biosensor composed of Au nanoparticle/$TiO_2$ nanoparticle/carbon nanotube, which can simultaneously detect AA, UA, DA and Trp with high sensitivity [19]. Priyatharshni et al. [20] determined AA, DA and UA simultaneously in pH 4 using an $ABO_3$-type $LaCoO_3$ chalcogenide material with high sensitivity for all three substances. Wang et al. modified carbon dots on GCE by drop coating and successfully assembled a biosensor capable of distinguishing between DA, UA, Trp and TP [21]. The detection of their precursors or metabolites of these analytes, such as phenylalanine [22,23], tyrosine [24,25], le xanthine [26,27] and serotonin [28,29], were also reported.

Carbon nano-onions (CNOs) are an important class of zero-dimensional materials with a large specific surface area, good electrical conductivity, good thermal and a closed and stable π–π conjugated structure. As a result, CNOs are expected to be widely used in optoelectronic, magnetic and wear-reducing material [30], especially in electrochemistry, as they have been reported to have promising applications as electrode materials, catalysts and biosensor building blocks in electrochemistry [31–33]. CNOs can promote electron transfer rates better than carbon nanotubes and show good electrocatalytic activity for some biomolecules [34,35]. However, studies on the application of CNOs as electrode modification materials in electrochemical sensors are relatively scarce and, therefore, have a very large scope for exploration. Ozoemena group investigated the electrochemical sensing of DA using onion-like carbons [36]. Plonska-Brzezinska and Echegoyen et al. determined DA in the presence of UA and AA using a carbon nano-onion and poly(diallyldimethylammonium chloride) composite [37]. To our knowledge, there is no report on the simultaneous detection of DA, UA, Trp and TP utilizing CNOs. As a result, the use of CNOs on modified electrodes as electrochemical sensors is of great importance.

In this work, structurally complete and high-purity CNOs were prepared by an organic electrochemical synthesis method, which had the advantages of mild reaction conditions, high product purity, easy operation and low environmental pollution [38]. The modified CNOs/GCE on the surface of glassy carbon electrodes were used for electrochemical studies of DA, UA, Trp and TP. Figure 1 is a schematic diagram of the CNOs/GCE analysis and calibration process. The results show that the CNOs/GCE sensor has excellent stability and can be used for real sample detection. This expands its application in the field of electrochemistry and has important research significance.

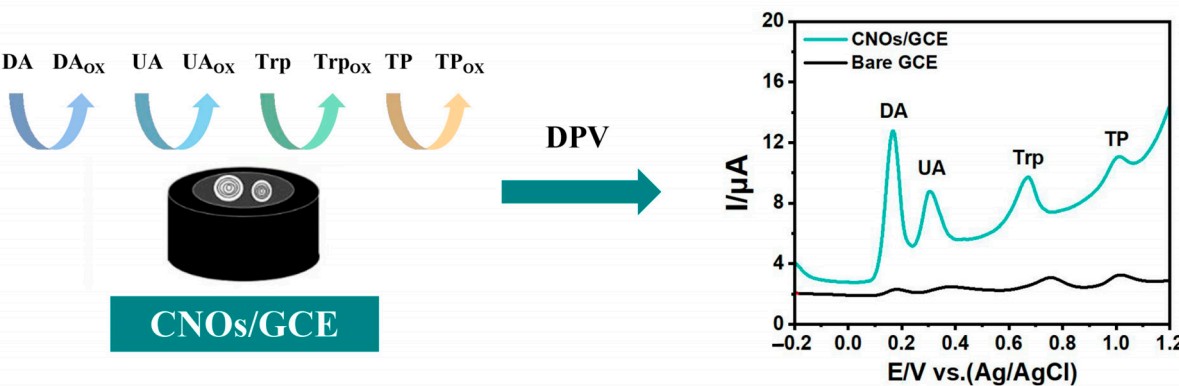

**Figure 1.** Schematic diagram of the modification and analysis procedure of CNOs/GCE for the simultaneous determination of DA, UA, Trp and TP.

## 2. Experimental

### 2.1. Materials and Reagents

Toluene, sulfuric acid and hydrogen peroxide (aqueous solution) were purchased from Xilong Chemical Co., Ltd. (Tianjin, China). Acetonitrile was from Thermo Fisher Scientific Co., Ltd. (Waltham, MA, USA). Tetrabutylammonium perchlorate (TBAP), benzene and 1,2-dimethylbenzene were from Sigma-Aldrich (Saint Louis, MO, USA). DA and UA were from Sigma-Aldrich LLC. Trp was from Bio Basic (Toronto, ON, Canada). TP was from Aladdin Industries (Shanghai, China). Phosphate buffer solution (PBS, 0.1 M) was prepared using $KH_2PO_4$ and $Na_2HPO_4$. Human serum samples were provided by the Affiliated Hospital of Renmin University. All chemicals were of analytical-reagent grade. All aqueous solutions were prepared with Milli-Q water (18.2 MΩ·cm).

### 2.2. Apparatus

Electrochemical analysis measurements such as cyclic voltammetry (CV) and differential pulse voltammetry (DPV) were carried out using a conventional three-electrode system on a CHI660D workstation in Shanghai Chenhua, China. The purification process was carried out using a muffle furnace (SX2-4-10N) to remove uncharacterized carbon and an ultrasonic generator (SB-2800) to ultrasonically disperse the fired samples, followed by a centrifuge (TG16-WS) to remove the platinum particles. The Raman spectra for the synthesized and purified CNOs were obtained by a Raman spectrometer (Xplora plus). The morphology of the CNOs was obtained by field emission high-resolution transmission electron microscopy (HRTEM) (JEM-2100F) and X-ray powder diffraction (XRD) (D8 Discover, Bruker (Massachusetts, MA, USA)).

### 2.3. Fabrication of the CNOs

The entire electrochemical synthesis processes were performed at a CHI660D electrochemical workstation. To avoid the influence of air, the experiments needed to be carried out not only at room temperature but also in a nitrogen environment, so the final choice was made to carry out the experiments in a glove box (VG1200/750TS, Vigor (Suzhou, China)). Platinum plate electrodes (30 × 30 × 0.1 mm) were used for both working and counter electrodes, and an Ag/AgCl electrode with a porous ceramic tube that prevents silver nanoparticles from entering the system was used as the reference electrode. Synthesize CNOs using 30 mL acetonitrile, 0.1 M TBAP and 5 mM toluene in a maximum capacity cell of 50 mL. The breakdown potential was 2.2 V to −1.6 V, the breakdown time was 10 s, z and the number of cycles was 2000 times [38].

The impurities in the CNOs prepared by the above method mainly contained amorphous carbon and Pt nanoparticles. After scraping the product off the electrode, the product was first rinsed with acetonitrile to remove TBAP, followed by removing amorphous carbon in a muffle furnace for 10 h. An abrupt reduction in sample mass could be observed in this

step. The heat-treated samples were dispersed in CHP for sonication and then centrifuged at a gradient of 6000 r/min to 10,000 r/min for 20 min, with CNOs collected at 10,000 r/min. The final CNO dispersion was prepared for subsequent experiments by centrifugation of a mixture of ethanol and water (ethanol/water = 1:1) [38].

### 2.4. Fabrication of the CNOs/GCE

2.0 mg of purified CNOs were dispersed with the aid of ultrasonic agitation in 20 mL of N,N-dimethylformamide (DMF) to give a 0.1 mg/mL black suspension. The GC electrode was first abraded with emery paper (No. 1500) and polished with 0.3 μm alumina slurry, then washed ultrasonically in distilled water and ethanol, respectively. Then, the GC electrode was coated by gradually casting 10 μL of the CNO suspension prepared above and then dried under an infrared lamp.

## 3. Results and Discussion

### 3.1. Surface Characterization of the CNOs

Figure 2A,B show TEM images of CNOs. The hollow polyhedron morphology of CNOs was observed by TEM, and the distribution was relatively uniform, containing abundant nanoparticles. The distribution of the outer diameter (outer size) of nanoparticles was about 5–10 nm, and the inner diameter was about 2–3 nm. There were 6–8 unequal layers evenly distributed between the inner shell layers, and the layer spacing was approximately equal to the graphite layer spacing of 0.336 nm. At the same time, the carbon nano-onion molecules also appeared to "adhere" to each other, which might be due to the introduction of surfactants during the separation and purification, resulting in a serious aggregation phenomenon. Some nanoparticles had edge defects and a non-spherical polygon state.

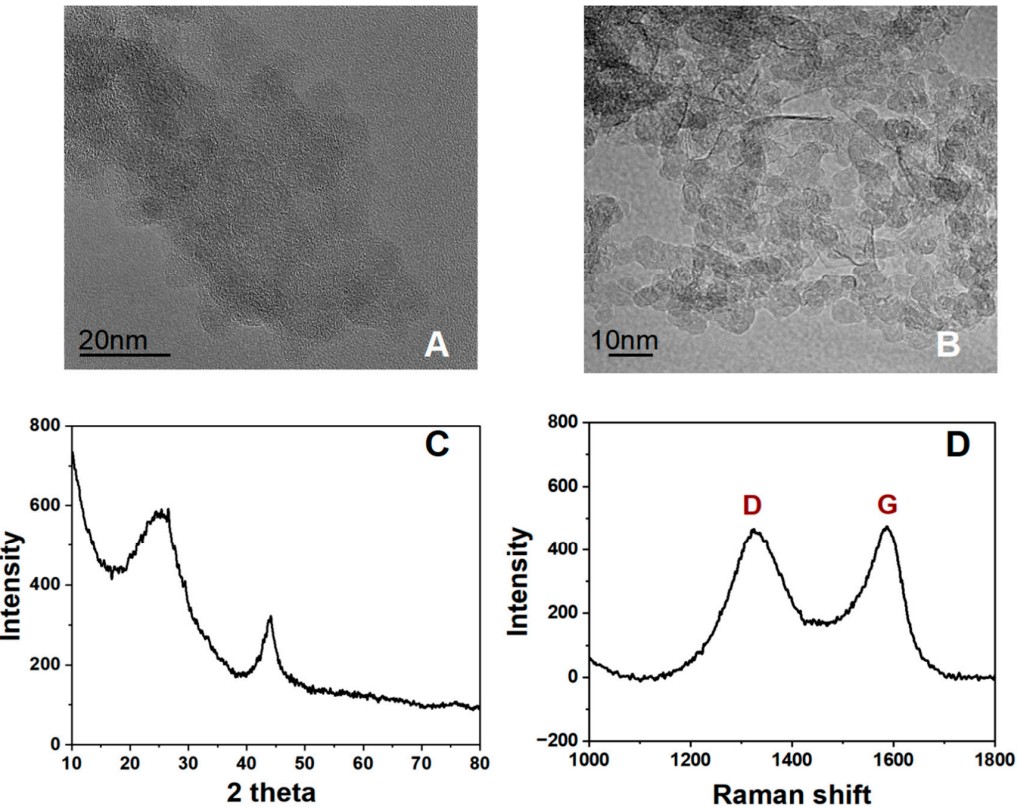

**Figure 2.** (**A**,**B**) TEM images of CNOs; (**C**) the XRD of the CNOs; (**D**) the Raman shift of the CNOs.

The X-ray diffraction technique was particularly suitable for the analysis of single and polycrystalline structures. Figure 2C shows the characterization results of XRD of CNOs produced by electrochemical synthesis. CNOs showed sharp diffraction peaks around

$2\theta = 26°$ and $2\theta = 44°$, close to those of graphite [002] at $26.4°$ on the crystal plane and graphite [101] at $44.6°$ on the crystal plane, where the layer spacing was altered due to the graphite layers curling to form a carbon onion and the tensile stress between the layers changing [39,40]. The more pointed peak shape and narrower half-peak width suggested that the nanoparticles in CNOs had a high degree of graphitization, corresponding to $sp^2$-hybridized carbon atoms.

Raman spectroscopy is a commonly used technique for characterizing the structure of carbon materials. Figure 2D shows the Raman spectrum of CNOs between 1200 and 1800 $cm^{-1}$, from which it could be seen that the results showed strong and clear G (graphitic) and D (disorder) bands, similar to those reported in the literature for CNOs obtained by nanodiamond powder heated to 1650 °C and annealing at room temperature [41], indicating that the CNOs were very small in size. The Raman spectral peak G mode of CNOs (at 1575 $cm^{-1}$) had a narrower peak width and higher intensity, thus indicating that the CNOs had a chemical structure similar to that of graphitic carbon atomic layers, where the particle structure was well-oriented and crystalline, which was highly consistent with the results obtained from previous XRD tests. Also visible in the figure is the strong spectral peak D modeled at 1359 $cm^{-1}$. This is about the same intensity as the G mode, indicating that CNOs contain more graphitized carbon onions but also more structural defects [42].

This was mainly due to the uneven size distribution of the carbon onion and the introduction of pentagons into the hexagonal graphite layer, which increased the spheroidal bond length and induced tension within the crystal, resulting in a frequency shift of the Raman scattering peak relative to 1582 $cm^{-1}$. The bending of carbon atomic layers in graphite crystals inevitably leads to defects or layer defects, destroying the integrity of the six-membered carbon atom ring network and generating D-mode near 1346 $cm^{-1}$ [43].

Therefore, the XRD and Raman test results indicated that the degree of graphitization and orderliness of the nanocarbon material products synthesized by electrochemical synthesis was high, which facilitated the study of their properties.

### 3.2. Electrochemical Reactivity of the CNOs/GCE Electrode

Electrochemical impedance spectroscopy is a common method for studying electron transfer between electrodes. The larger the diameter of the curved semicircle, the greater the transfer resistance ($R_{ct}$) of the electrode surface. Figure 3A shows the EIS spectra of GCE (black) and CNOs/GCE (red) in 5 mM $[Fe(CN)_6]^{3-/4-}$ and 0.1 M KCl solutions. The GCE had low arcing [44,45] in the high-frequency range and a charge transfer resistance ($R_{ct}$) of approximately 293 $\Omega$. After modifying CNOs on the surface of GCE, the Nyquist diagram of the modified electrode was changed, and the $R_{ct}$ value of the modified electrode was increased to 327.9 $\Omega$, indicating the modification of CNOs on GCE, and the increase of CNOs/GCE $R_{ct}$ should be related to the formation of the CNOs layer [46]. $R_s$ stood for the resistance of the solution, $C_d$ represented double-layer capacitance, $R_{ct}$ represented the interfacial charge transfer reaction, $Z_w$ stood for Warburg diffusion and $C_p$ and $R_p$ represented the dielectric properties and the resistance of the inner layer of the electrode.

To confirm that CNOs can facilitate electron transfer better than GCE, their electrochemical behavior at both electrodes was characterized using $[Fe(CN)_6]^{3-}$ as a redox probe. Figure 3B shows CV diagrams of CNOs/GCE and GCE in 0.1 mM $[Fe(CN)_6]^{3-}$ a solution containing 0.1 M KCl. The $I_{pa}$ and $I_{pc}$ at GCE were 123 $\mu$A and 124 $\mu$A, while the $I_{pa}$ and $I_{pc}$ at CNOs/GCE were 260 $\mu$A and 278 $\mu$A. Furthermore, the potential difference ($\Delta E_p$) of the redox peak on the GCE was 89 mV, while the $\Delta E_p$ on the CNOs/GCE electrode was reduced to 83 mV. In addition, the electrochemically active surface area ($A_e$) of the electrode was calculated using the Randles–Sevcik equation:

$$I_{pa} = 2.69 \times 10^5 A_e n^{3/2} D_0^{1/2} C_0 V^{1/2}$$

In the formula, n is the electron transfer number, $D_0$ is the diffusion coefficient of $[Fe(CN)_6]^{3-}$ in an aqueous solution ($6.3 \times 10^{-6}$ $cm^2/s$), V is the sweep speed, and $C_0$ is the volume concentration. The calculated $A_e$ values for GCE and CNOs/GCE were 0.36 and

0.77 cm$^2$, respectively, proving that the electrode modified with carbon nano-onion had significantly improved electron transport capacity and enhanced electrochemical response.

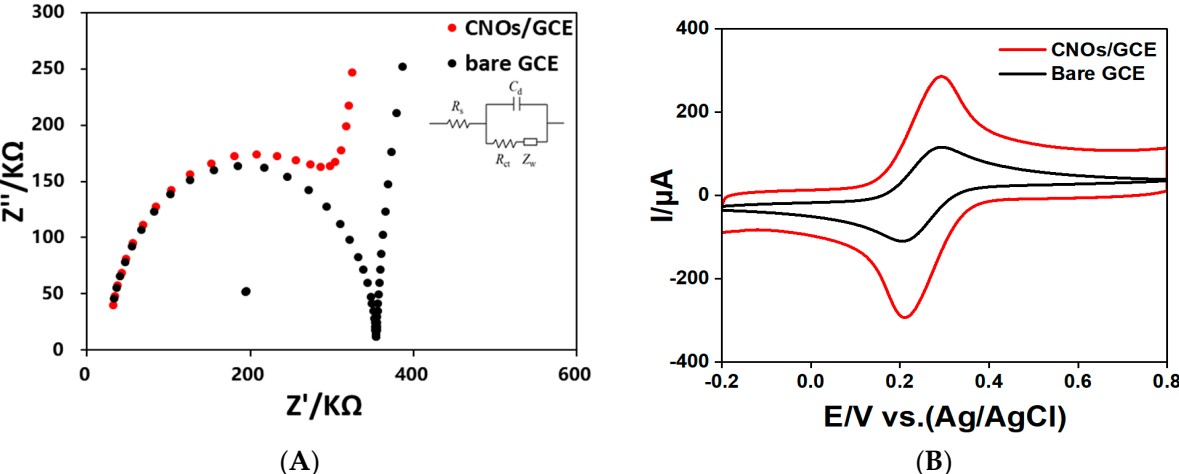

(A)                      (B)

**Figure 3.** (**A**) Nyquist plots of CNOs/GCE and GCE at 0.1 mM [Fe(CN)$_6$]$^{3-/4-}$ in 0.1 M KCl; (**B**) CV obtained at the CNOs/GCE and GCE at 0.1 mM [Fe(CN)$_6$]$^{3-}$ in 0.1 M KCl.

### 3.3. Electrochemical Performance of CNOs/GCE for DA, UA, Trp and TP

To compare the electrochemical performance of GCE and CNOs/GCE, DA, UA, Trp and TP were simultaneously detected by differential pulse voltammetry (DPV). Figure 4 shows the DPV plots of the four substances on GCE and CNOs/GCE, respectively. On GCE, distinct oxidation peaks were observed for DA, UA, Trp and TP with peak potentials of 0.184 V, 0.34 V, 0.756 V and 1.016 V, respectively, while on CNOs/GCE, four distinct oxidation peaks were observed for DA, UA, Trp and TP at 0.164 V, 0.312 V, 0.672 V and 1.008 V, respectively. The oxidation peak potentials were all negatively shifted. The peak potential separations were 148 mV for DA and UA, 360 mV for UA and Trp, and 336 mV for Trp and TP. The peak oxidation currents of DA, UA, Trp and TP on CNOs/GCE increased significantly by a factor of about 10, 9, 3 and 2, respectively. This indicated that CNOs/GCE had a sufficient resolution for the potential difference of the mixed four substances to determine the mixed four substances simultaneously.

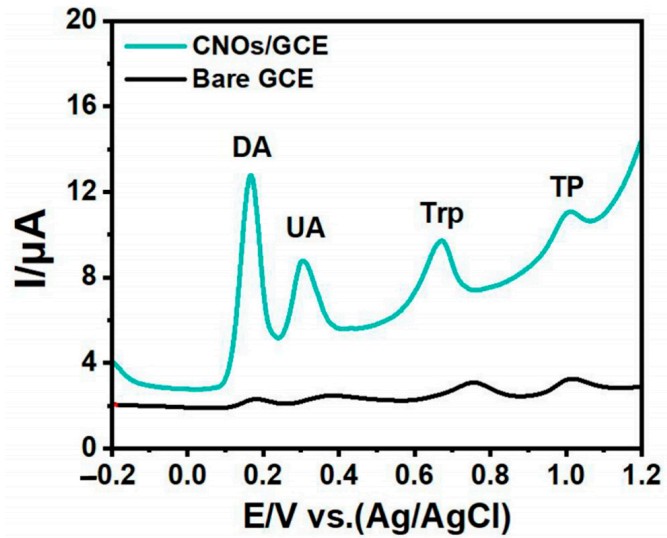

**Figure 4.** DPV was obtained at GCE and CNOs/GCE in 0.1 M PBS (pH 7.0) of 10 μM DA, 20 μM UA, 20 μM Trp and 20 μM TP. Scan rate: 50 mV/s.

*3.4. Optimization of Parameters*

3.4.1. Investigation of the pH Effects

Optimize the pH of the buffer and observe the effects of different pH (3.0, 4.0, 5.0, 6.0, 7.0 and 8.0) solutions on the oxidation peak currents and peak potential differences of DA, UA, Trp and TPon CNOs/GCE using DPV method for analysis. The result is shown in Figure 5. The DPV plots for DA, UA, Trp and TP are given in Figure 5A–D, respectively, where the corresponding oxidation peak potentials were negatively linear with pH. The linear regression equation corresponding to DA was $E_p$ (V) = −0.0610 pH + 0.5893 (R = 0.9996), the linear regression equation corresponding to UA was $E_p$ (V) = −0.0575 pH + 3.457 (R = 0.9926), with the slope of the linear regression line for DA and UA closed to 59 mV·pH$^{-1}$ (theoretical value), which indicated that the electrooxidation reaction involved the same amount of proton and electron transfer. The linear regression equation corresponding to Trp was $E_p$ (V) = −0.0391 pH + 7.911 (R = 0.9963), and the linear regression equation corresponding to TP was $E_p$ (V) = −0.0477 pH + 1.324 (R = 0.9918). The slopes of the linear regression lines for Trp and TP were both approximately 1/2 the theoretical value, indicating that the number of protons transferred during the reaction was 1/2 the number of electrons transferred.

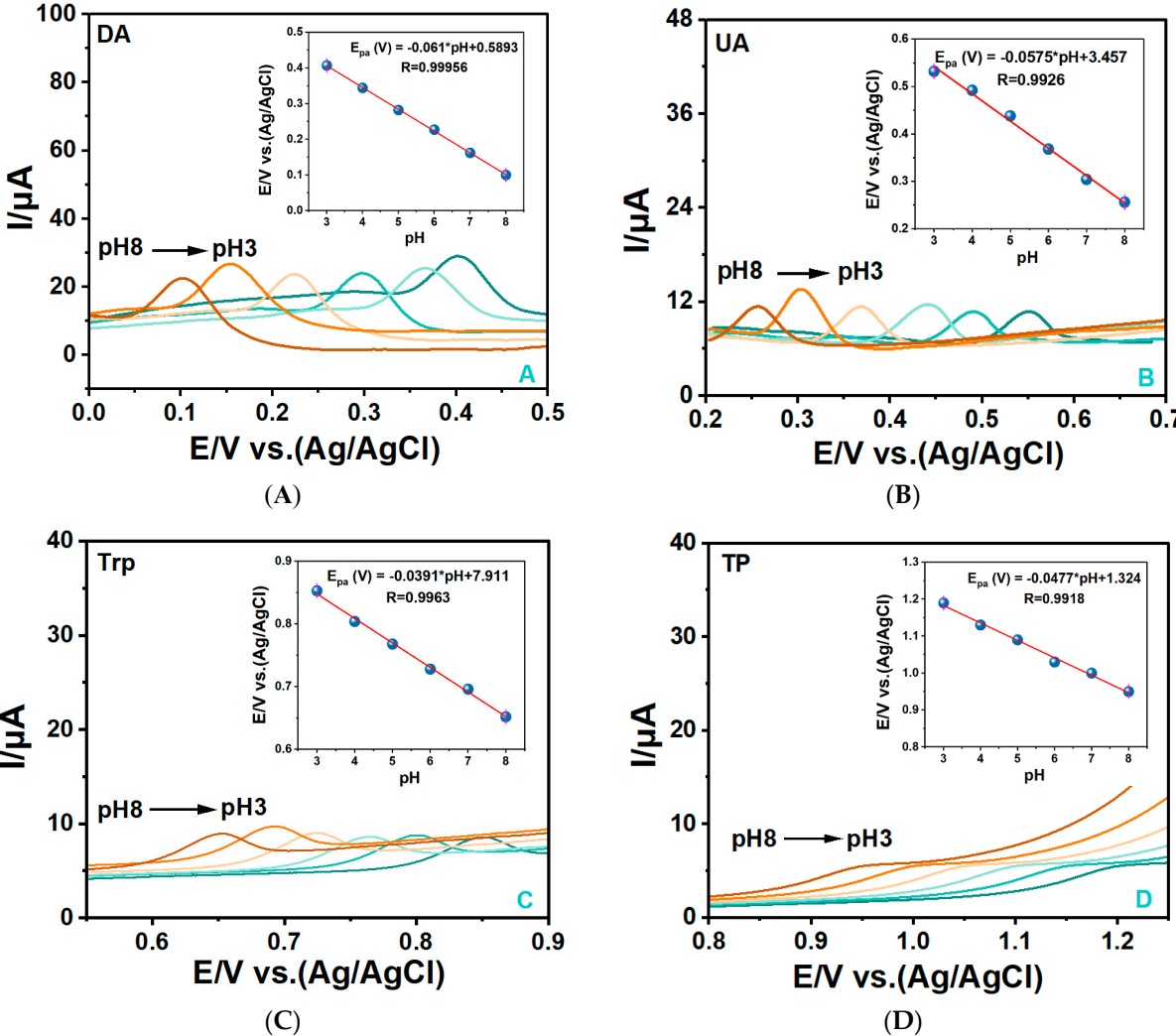

**Figure 5.** DPVs of 10 μM DA (**A**), 40 μM UA (**B**), 40 μM Trp (**C**) and 40 μM TP (**D**) on CNOs/GCE in PBS with different pH values (3.0–8.0), Scan rate: 100 mV/s; Insets: The linear relationships of PH and peak potential at CNOs/GCE of DA, UA, Trp and TP.

Figure 6A shows the variation of oxidation peak currents for DA, UA, Trp and TP as affected by pH. It could be seen that the largest oxidation peak currents appeared at pH 7 for DA, UA and TP and pH 4 for Trp, respectively. Figure 6B depicts the variation of oxidation peak difference between DA, UA, Trp and TP as affected by pH, where the UA-DA oxidation peak separation increased slowly with increasing pH, with a minimum value at pH 3 and a larger value at pH 7 and 8. The Trp-UA oxidation peak separation tended to decrease and then increase with increasing pH, with a minimum value at pH 4 and a larger value at pH 7 and 8. The TP-Trp oxidation peak separation showed a slight decrease with increasing pH, with larger values at pH 3 and 4. The TP-DA oxidation peak separation changed little at pH 3–8.

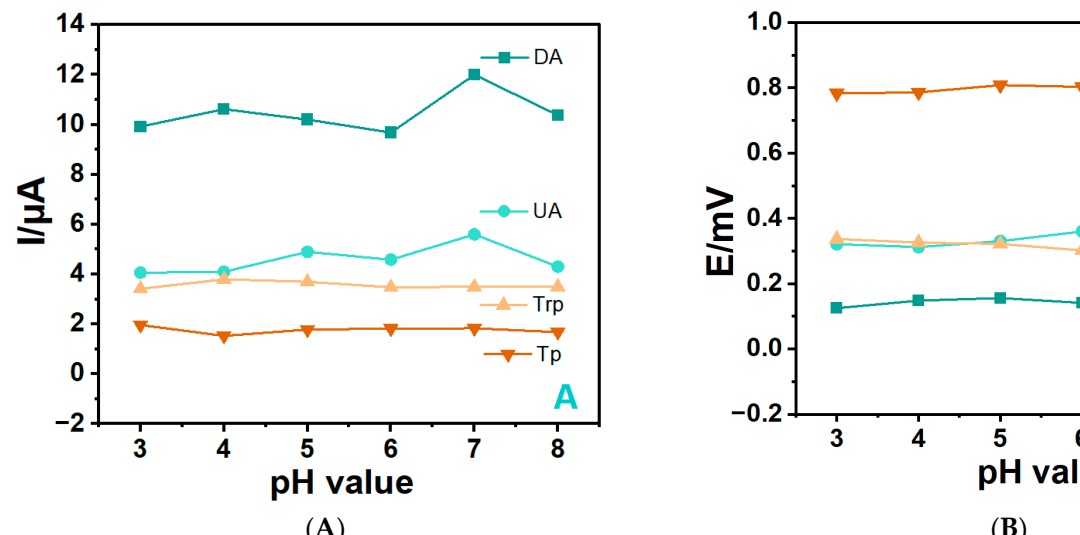

**Figure 6.** Effects of pH values on peak current (**A**) and peak potential (**B**) of 10 μM/L DA, 20 μM/L UA, 20 μM/L Trp and 20 μM/L TP on CNOs/GCE.

Therefore, pH 7 showed that the oxidation peak potential separation of UA-DA, Trp-UA and TP-DA was larger, and the corresponding response currents of DA, UA and TP oxidation peaks were higher, and this pH value was comparable to the human physiological pH environment, so this experiment selected pH 7 as the best detection condition.

3.4.2. The Effect of Scan Rates

The effect of scan rate (50–500 mV/s) on the peak anodic current of DA, UA, Trp and TP detected by CNOs/GCE was investigated using cyclic voltammetry, as shown in Figure 7. The peak oxidation currents of the four substances were linearly related to the square root of the scanning rate, indicating that the electrooxidation reactions of the four substances on CNOs/GCE were diffusion-controlled processes, following linear regression equations:

$$I_{pa}(DA) = 1.0295v^{1/2} - 0.1185, R = 0.9991$$

$$I_{pa}(UA) = 0.5383v^{1/2} - 0.0726, R = 0.9906$$

$$I_{pa}(Trp) = 0.3028v^{1/2} - 0.7501, R = 0.9920$$

$$I_{pa}(TP) = 0.1844v^{1/2} + 0.4119, R = 0.9926$$

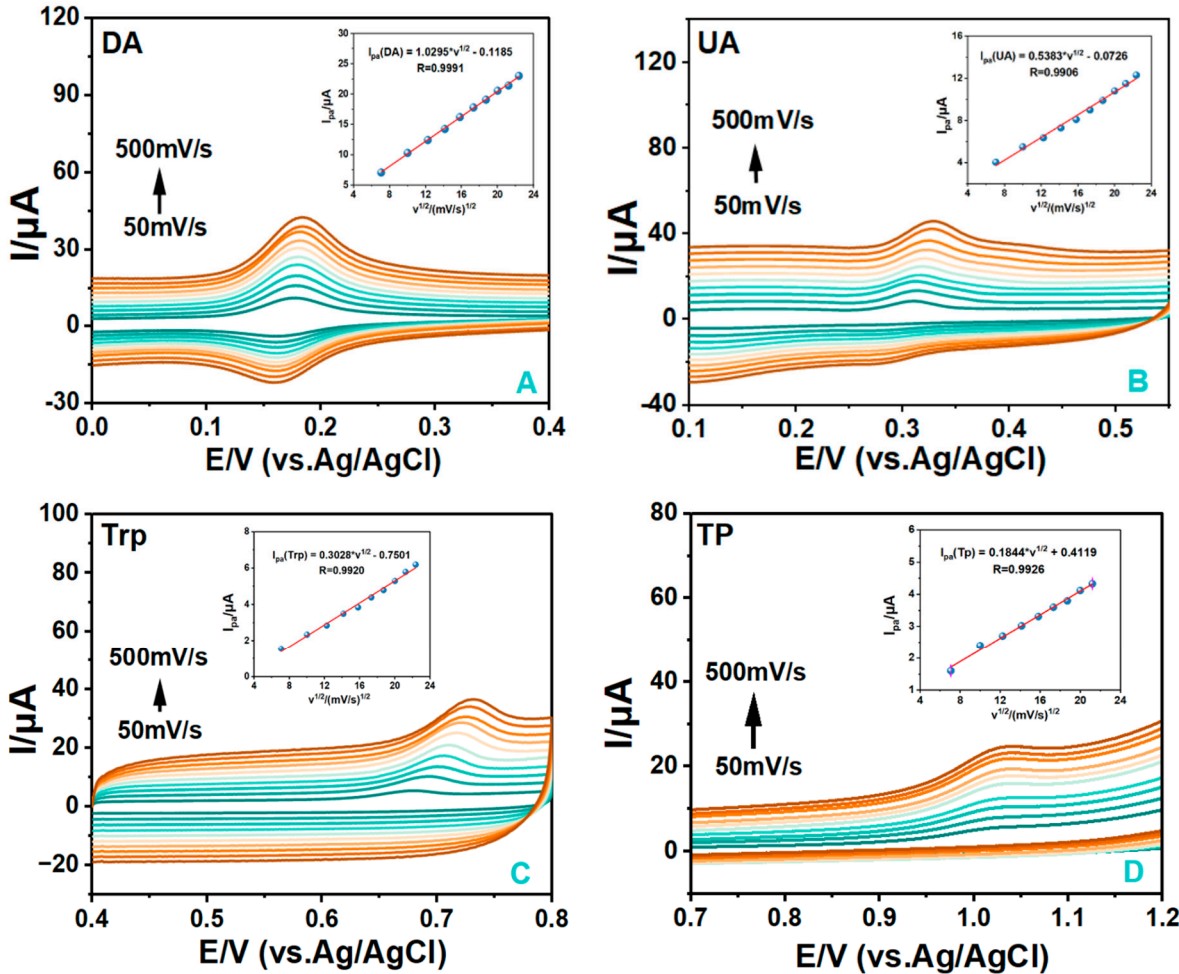

**Figure 7.** CVs of 10 µM DA (**A**), 40 µM UA (**B**), 40 µM Trp (**C**) and 40 µM TP (**D**) on CNOs/GCE in PBS (pH 7.0) at different scan rates (50–500 mV/s); Insets: the linear relationships of $v^{1/2}$ (mV/s) vs. $I_{pa}$ (µA) for DA, UA, Trp and TP on CNOs/GCE.

### 3.4.3. Individual and Simultaneous Determination of DA, UA, Trp and TP

The different DPV responses of DA, UA, Trp and TP were recorded under the condition of 0.1 M PBS (pH 7.0), where the concentration of one substance was varied while the concentrations of the other three substances remained unchanged (Figure 8A–D). The results showed that when the four substances were tested simultaneously, the peak oxidation currents of the substances with individually varying concentrations increased linearly with increasing concentration, whereas the peak currents of the three remaining substances whose concentrations were kept constant remained almost unchanged. Based on the linear results, the regression equations for DA, UA, Trp and TP were as follows:

$$I_{DA}\ (\mu M) = 0.0671 + 0.9963\ C_{DA}\ (\mu M),\ R = 0.9938$$

$$I_{UA}(\mu A) = 0.0364 + 0.1078\ C_{UA}\ (\mu M),\ R = 0.9988$$

$$I_{Trp}(\mu A) = 0.0070 + 0.0802\ C_{Trp}\ (\mu M),\ R = 0.9977$$

$$I_{TP}(\mu A) = 0.1069 + 0.0469\ C_{TP}\ (\mu M),\ R = 0.9981$$

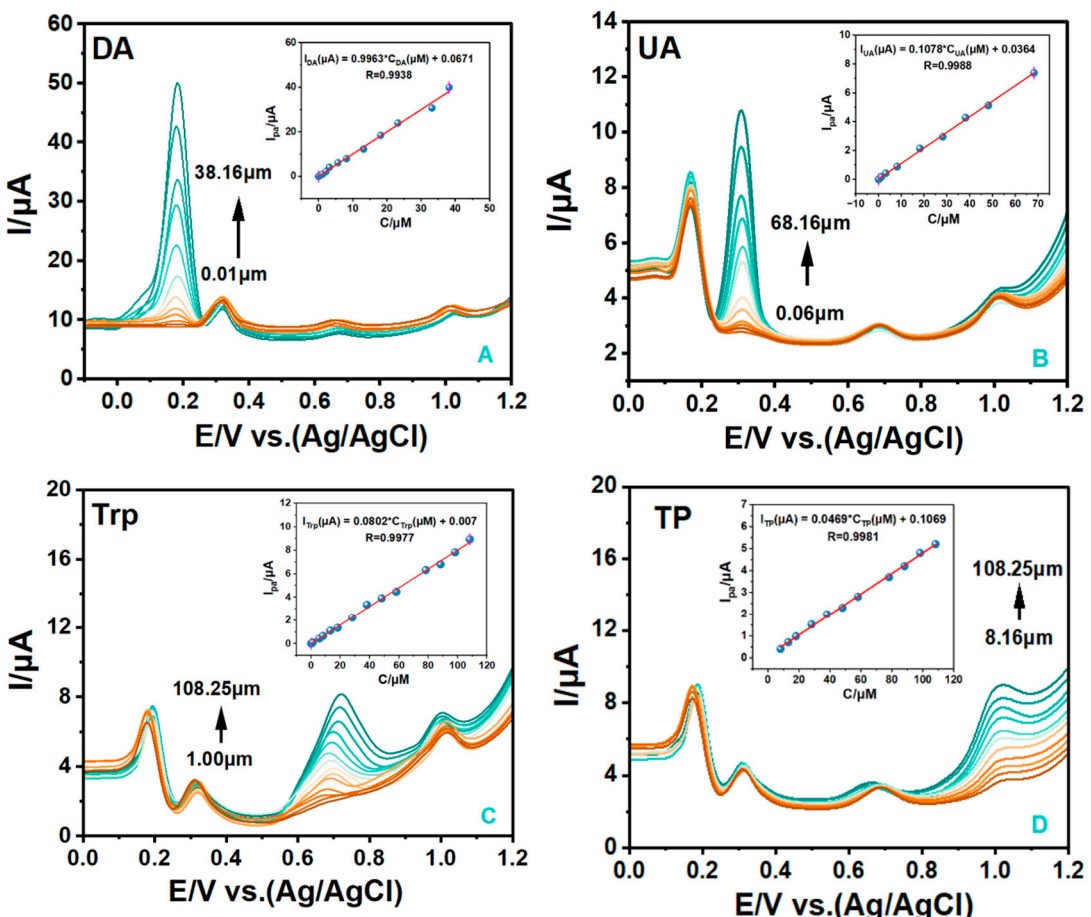

**Figure 8.** DPV under CNOs/GCE in 0.1 M PBS (pH 7.0) containing (**A**) 20 μM UA, 20 μM Trp and 20 μM TP and different concentrations of DA (0.01 μM–38.16 μM); (**B**) 10 μM DA, 20 μM Trp and 20 μM TP in 0.1 M PBS (pH 7.0) and different concentrations of UA (0.06 μM–68.16 μM) in DPV under CNOs/GCE; (**C**) 10 μM DA, 20 μM UA and 20 μM TP in 0.1 M PBS (pH 7.0) and different concentrations of Trp (1.00 μM–108.25 μM) in DPV under CNOs/GCE; (**D**) 10 μM DA, 20 μM UA and 20 μM Trp in 0.1 M PBS (pH 7.0) and different concentrations of TP (8.16 μM–108.25 μM) at DPV under CNOs/GCE. Insets: the calibration plots of DA, UA, Trp and TP peak currents versus concentration, respectively.

The results showed that CNOs/GCE was resistant to interference in the detection system and could detect DA, UA, Trp and TP simultaneously on CNOs/GCE. The detection limit of DA was 0.0039 μM/L (S/N = 3), and the relative standard deviation was ±1.4% (*n* = 10). Similarly, the detection limits of UA, Trp and TP were 0.0087 μM/L (S/N = 3), 0.18 μM/L (S/N = 3) and 0.35 μM/L (S/N = 3), respectively, with RSD of ±1.05% (*n* = 10), ±2.00 (n = 10) and ±1.33% (*n* = 10), respectively. Table 1 lists the sensing performances of the CNOs/GCE compared with other related sensors reported in recent literature. Actually, the CNOs/GCE showed better performances than other electrodes, especially the wide linear ranges and low detection limits.

**Table 1.** Comparison of the CNOs/GCE sensor with other modified electrodes for the determination of DA, UA, Trp and TP.

| Biosensors | Linear Range (μM) | | | | Detection Limit (μM) | | | | Ref. |
|---|---|---|---|---|---|---|---|---|---|
| | DA | UA | Trp | TP | DA | UA | Trp | TP | |
| (Au-PDNs)/GCE | 1–160 160–350 | 1–120 120–350 | 1.0–160 160–280 | \ | 0.0001 | 0.0001 | 0.0001 | \ | [47] |
| Ni-ZIF-8/N S-CNTs/CS/GCE | 8–500 | 1–600 | 1–600 | \ | 0.93 | 0.41 | 0.69 | \ | [48] |

**Table 1.** *Cont.*

| Biosensors | Linear Range (µM) | | | | Detection Limit (µM) | | | | Ref. |
|---|---|---|---|---|---|---|---|---|---|
| | DA | UA | Trp | TP | DA | UA | Trp | TP | |
| poly(CTAB)/GCE | 0.5–1000 | 1–1000 | 1–1000 | 0.5–1000 | 0.11 | 0.33 | 0.44 | 0.11 | [49] |
| pPABSA/GCE | \ | \ | \ | 0.9–100 | \ | \ | \ | 7.02 | [50] |
| β-CD/CQDs/GCE | 4–220 | 0.3–200 | 5–270 | \ | 0.14 | 0.01 | 0.16 | \ | [18] |
| LMC/Nafion/GCE | \ | \ | \ | 0.8–180.0 | \ | \ | \ | 0.37 | [51] |
| CDs/GCE | 0.5–50 | 3–28.5 28.5–75 | 1–65 | 10–200 | 0.007 | 0.011 | 0.11 | 0.33 | [21] |
| Nitrogen-doped graphene/GCE | 0.5–170 | 0.1–20 | \ | \ | 0.25 | 0.045 | \ | \ | [13] |
| NCCNPs/GCE | 0.002–0.0695 | 0.005–0.192 | \ | \ | 0.34 | 0.98 | \ | \ | [52] |
| 1,4-BBFT/ Carbon paste | \ | \ | \ | 0.06–700 | \ | \ | \ | 0.012 | [7] |
| PAMT/AuNPs/TiO$_2$@CuO-B/RGO/GCE | \ | 0.0005–10 | \ | 0.001–10 | \ | 0.00018 | \ | 0.00036 | [53] |
| Hand-in-hand RNA nanowire | \ | \ | \ | 0.5–70 | \ | \ | \ | 0.05 | [54] |
| ZIF-8@CoTA/CPE | 0.02–0.44 | 0.02–0.44 | 0.02–0.44 | \ | 0.0012 | 0.0067 | 0.0051 | \ | [55] |
| MoO$_3$@B/N-PC | 0.079–104.1 | 0.086–102.5 | \ | \ | 0.069 | 0.078 | \ | \ | [56] |
| CoTGPc/GCE | 2–10 | 2–10 | \ | \ | 0.03 | 0.1 | \ | \ | [57] |
| Co$_x$Ni$_{1-x}$(OH)$_2$/G | 0.1–750 | 0.25–925 | \ | \ | 0.043 | 0.097 | \ | \ | [58] |
| CNOs/GCE | 0.01–38.16 | 0.06–68.16 | 1.00–108.25 | 5.16–108.25 | 0.0039 | 0.0087 | 0.18 | 0.35 | This work |

*3.5. Interference Studies*

To investigate the anti-interference ability of CNOs/GCE as a DA, UA, Trp and TP sensor, the amperometric responses of CNOs/GCE were continuously measured in a solution containing 5 µM DA, 20 µM UA, 20 µM Trp and 20 µM TP. Prepare a mixed solution in 0.1 M PBS at pH 7.0 containing potentially interfering species, including 50 mM Cl$^-$, SO$_4^{2-}$, K$^+$ and 100 mM Na$^+$ inorganic ions, 5 mM glycine, L-lysine, L-valine, adenine and 50 µM ascorbic acid (Figure 9). It can be seen that the addition of the interfering substance did not cause a significant change in the current response to it. And this stable current signal was somewhat diminished during the experiment as the interfering substance solutions were continuously added to increase the overall solution volume, causing small decreases in DA, UA, Trp and TP concentrations. Nevertheless, upon re-addition of 5 µM DA, 20 µM UA, 20 µM Trp and 20 µM TP, the actual response was prominent. These results indicated that the CNOs/GCE has a certain degree of anti-interference ability in the detection system, which provides prospects for the potential application of CNOs/GCE in the field of non-enzymatic detection.

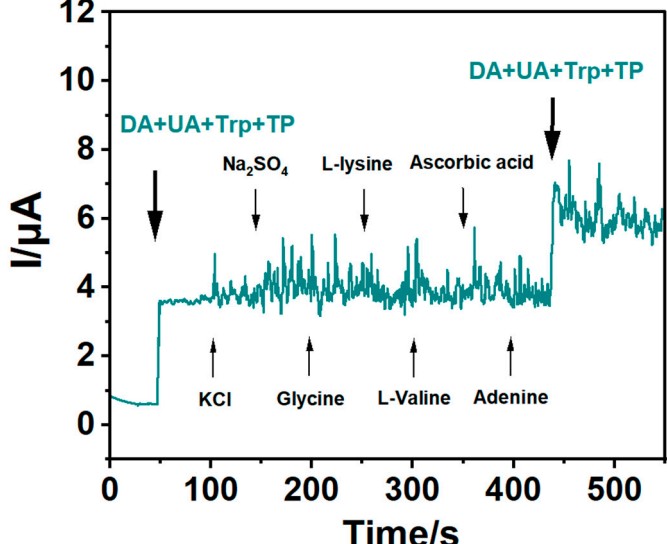

**Figure 9.** Interference test of CNOs/GCE in 0.1M PBS (pH 7.0) solution in the working potential of 0.81 V with 5 µM DA, 20 µM UA, 20 µM Trp, 20 µM TP and other interfering substances (50 mM Cl$^-$, SO4$^{2-}$, K$^+$ and 100 mM Na$^+$ inorganic ions, 5 mM glycine (Gly), L-lysine (L-Lys), L-valine (L-val), adenine and 50 µM ascorbic acid (AA)).

### 3.6. Reproducibility, Repeatability and Stability Studies

The stability of CNOs/GCE was assessed every 10 days in PBS solution (pH 7.0) containing 20 μM DA and 30 μM UA, Trp and TP (Figure 10). After 30 days, the current response of the modified electrode to DA, UA, Trp and TP was 91.56%, 95.84%, 93.91% and 93.24% of the initial values, respectively, with the placed electrode. The performance of the placed electrodes decreased but maintained the detection status, indicating that the electrodes have good stability.

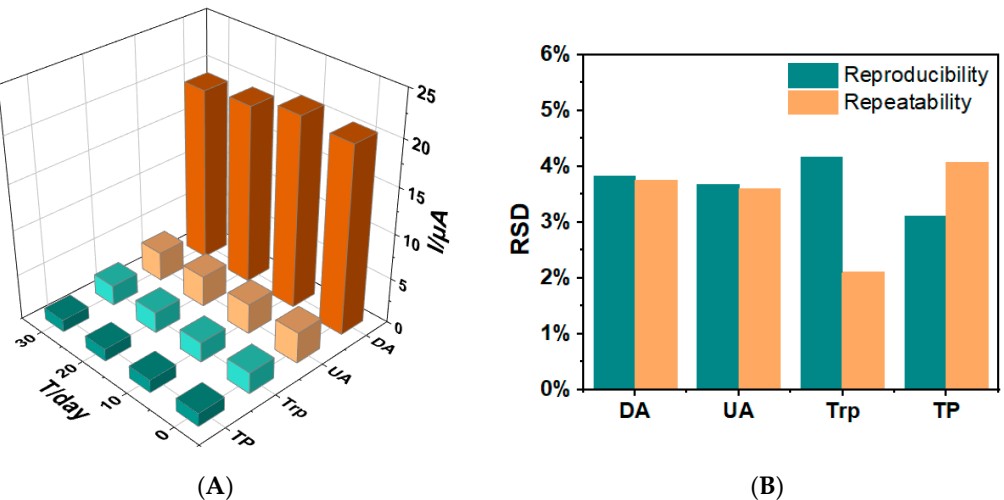

(**A**)                (**B**)

**Figure 10.** (**A**) Electrochemical responses of CNOs/GCE to 20 μM DA, 30 μM UA, Trp and TP every ten days. (**B**) RSD data for reproducibility and repeatability of CNOs/GCE for 30 μM DA, UA, Trp and TP.

Then, prepare 6 CNOs/GCEs by the same method and check the reproducibility in PBS (pH 7.0) containing 30 μM DA, UA, Trp and TP, respectively. The reproducibility relative standard deviations (RSD) of DA, UA, Trp and TP were 3.82%, 3.67%, 4.16% and 3.10%, respectively.

On the other hand, the same sensor was used for nine consecutive measurements at the same time. The RSDs of DA, UA, Trp and TP were 3.75%, 3.59%, 2.1% and 4.07%, respectively, with good reproducibility. Studies have shown that CNOs/GCE had good reproducibility, repeatability and stability in the simultaneous determination of DA, UA, Trp and TP.

### 3.7. Real Samples Analysis

The standard addition method was applied in the environment of human serum to further evaluate the effectiveness of CNOs/GCE for the detection of DA, UA, Trp and TP. Also, the DPV technology was used in the process. Before the measurement, dilute 1.0 mL of the original serum solution with 0.1 M PBS (pH 7.0) to 10.0 mL, and add the recoveries of various DA, UA, Trp and TP standard solutions into the blood sample solution (Table 2). The results showed that CNOs/GCE responded well to real samples of DA, UA, Trp and TP with satisfactory recoveries, confirming the good catalytic activity of CNOs/GCE for the clinical diagnosis of DA, UA, Trp and TP.

**Table 2.** Testing data of human serum.

| Sample | Serum (μM) | | | | Added (μM) | | | | Founded (μM) | | | | Recovery (%) | | | |
|---|---|---|---|---|---|---|---|---|---|---|---|---|---|---|---|---|
| | DA | UA | Trp | TP | DA | UA | Trp | TP | DA | UA | Trp | TP | DA | UA | Trp | TP |
| 1 | 0 | 10.93 | 2.59 | 0 | 3.5 | 20 | 6 | 35 | 3.31 | 29.7 | 8.05 | 33.61 | 94.57 | 96.02 | 93.71 | 96.03 |
| 2 | 0 | 12.81 | 3.24 | 0 | 4.5 | 25 | 12 | 55 | 4.27 | 35.81 | 14.52 | 54.98 | 94.89 | 94.71 | 95.28 | 99.96 |
| 3 | 0 | 10.66 | 1.13 | 0 | 5.5 | 30 | 18 | 75 | 5.33 | 38.74 | 18.47 | 74.19 | 96.91 | 95.28 | 96.55 | 98.92 |

## 4. Conclusions

In this study, carbon nano-onions (CNOs) were fabricated via a stepwise multipotential method and were dropped onto glassy carbon electrodes (GCEs) for immobilization. A novel simple and highly sensitive electrochemical biosensor was developed to simultaneously measure DA, UA, Trp and TP. The sensor has high sensitivity, low detection limit, good repeatability and good test stability. In addition, the CNOs/GCE sensor exhibits excellent anti-interference ability against a variety of common species and has potential applications in the determination of DA, UA, Trp and TP in human serum.

**Author Contributions:** Conceptualization, R.A.,T.M. and H.L.; methodology, R.A.; software, Z.L.; validation, R.A.; formal analysis, R.A.; investigation, R.A.; resources, R.A.; data curation, W.K. and Z.L.; writing—original draft preparation, R.A. and H.L.; writing—review and editing, R.A., T.M. and H.L.; visualization, R.A.; supervision, H.L.; project administration, R.A., T.M. and H.L.; funding acquisition, T.M. and H.L. All authors have read and agreed to the published version of the manuscript.

**Funding:** This study was supported by the National Natural Science Foundation of China (No.22073112).

**Institutional Review Board Statement:** Not applicable.

**Informed Consent Statement:** Not applicable.

**Data Availability Statement:** The data presented in this study are available on request from the corresponding author.

**Conflicts of Interest:** The authors declare no conflict of interest.

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
