# Peer review of "A Simple Strategy for the Simultaneous Determination of Dopamine, Uric Acid, L-Tryptophan and Theophylline Based on a Carbon Nano-Onions Modified Electrode"

_processes, doi:10.3390/pr11092547_

Round 1

Reviewer 1 Report

The manuscript provides an insightful examination into the characterization of Carbon Nano Onions (CNOs). The discussion is detailed, informative, and generally well-structured, but there are some points that require revision to enhance the clarity and coherence of the document.

1.     Excessive use of abbreviations;

2.     When referring to the figures in the manuscript, it would be more reader-friendly to include a brief description of each figure.

3.     Technical Jargon: While technical language is necessary for this type of work, it's important to ensure it doesn't hinder understanding for readers who might be slightly less familiar with the area. If possible, define or explain complex technical terms. For instance, clarify the meaning of terms like 'sp2-hybridized carbon atoms', 'D and G bands', or 'agglomeration phenomenon'.

4.     In some parts, the interpretation of results may not be very clear to the reader. For example, "The Raman spectral peak G mode of CNOs (at 1575 cm-1) had a narrower peak width and higher intensity, thus indicating that the CNOs had a chemical structure similar to that of graphitic carbon atomic layers." For an unfamiliar reader is difficult to follow.

5.     A contextual importance of such a highly sensitive electrochemical biosensor, considering existing detecting methods is needed.

By addressing these issues, the manuscript can be significantly improved in terms of readability and comprehension. This, in turn, will ensure the valuable findings are clearly communicated and understood by the readers.

Minor editing

Author Response

The manuscript provides an insightful examination into the characterization of Carbon Nano Onions (CNOs). The discussion is detailed, informative, and generally well-structured, but there are some points that require revision to enhance the clarity and coherence of the document.

  1. Excessive use of abbreviations;

Response: We are very sorry for this inconvenience as we have deleted some abbreviations in the revised manuscript.   

  1. When referring to the figures in the manuscript, it would be more reader-friendly to include a brief description of each figure.

Response:  Thank you very much for your kind suggestion. We have added the omitted descriptions in the revised manuscript.

  1. Technical Jargon: While technical language is necessary for this type of work, it's important to ensure it doesn't hinder understanding for readers who might be slightly less familiar with the area. If possible, define or explain complex technical terms. For instance, clarify the meaning of terms like 'sp2-hybridized carbon atoms', 'D and G bands', or 'agglomeration phenomenon'.

Response: Thank you very much for your kind suggestions. We added the description of D and G band in 3.1, and change the 'agglomeration phenomenon' to 'aggregation phenomenon'

  1. In some parts, the interpretation of results may not be very clear to the reader. For example, "The Raman spectral peak G mode of CNOs (at 1575 cm-1) had a narrower peak width and higher intensity, thus indicating that the CNOs had a chemical structure similar to that of graphitic carbon atomic layers." For an unfamiliar reader is difficult to follow.

Response: Thank you very much for pointing out this problem, we added reference 42 to help the reader to follow.

  1. A contextual importance of such a highly sensitive electrochemical biosensor, considering existing detecting methods is needed.

Response: Reference 36 and 37 were added in the revised manuscript as existing detecting methods.

By addressing these issues, the manuscript can be significantly improved in terms of readability and comprehension. This, in turn, will ensure the valuable findings are clearly communicated and understood by the readers.

Response: Thank you very much for your kind help.

Reviewer 2 Report

see file attacced

Minor editing of English language required

Author Response

Review comments

Manuscript processes-2536469 entitled “A Simple Strategy for the Simultaneous Determination of Do-2 pamine, Uric Acid, L-Tryptophan and Theophylline Based on a  Carbon Nano Onions Modified Electrode” presents an excellent work of simultaneous detection of analytes with applications, even in real samples. However I do suggest some changes before being accepted.

Response: Thank you very much for your positive comments to our manuscript, we have revised the manuscript according to your kind suggestions.

1) The introduction should be improved by broadening the discussion on the importance of the detection of these compounds also showing how in the literature there are works that seek to detect precursors or metabolites of these such as phenylalanine (10.1109/TIM.2023.3284027, 10.1002/elan.202200501, etc.), la tyrosine(10.1016/j.mtcomm.2023.106036,/10.1016/j.bios.2023.115360,etc), le xanthine(10.1007/s10800-023-01916-w, 10.1007/s00604-022-05601-1, etc) o serotonin (10.1016/j.bios.2022.114990, 10.1039/D2AY01627C, etc ). The discussion on carbon nanoonions should also be extended (10.1016/j.carbon.2015.10.089, etc.).

Response: Thank you very much for your kind suggestions. We have added these literatures and discussions in the revised manuscript.

2) I would emphasize more the novelty of this work. As the synthesis is not new, and also the use of carbon nano onions is already reported in the literature for the detection of dopamine (10.1016/j.electacta.2012.03.177, 10.1007/s12678-019-00520-x, etc). So even in the text where you say that no works in the literature specified better what you refer to so as not to seem misleading.

Response: We are very sorry for our mistake, and thank you very much for the literature. We have made corresponding changes in the revised manuscript.

3) Image 2a should be improved if possible.

Response: Image 2a was improved in the revised manuscript.

4) Paragraph 3.2 there are several inconsistencies. Indicative of having used [Fe(CN)6]3-/4-for eis and [Fe(CN)6]3- for CV why?

Response: As the excellent reversibility of [Fe(CN)6]3- or [Fe(CN)6]4-, the CV of [Fe(CN)6]3- would not change with or without [Fe(CN)6]4-.

5) Paragraph 3.2. The concentration of [Fe(CN)6]3-/4- in the text is indicated as 0,1 mM while in the figure caption is 5mM.

Response: We are very sorry for this mistake. We have changed “5 mM” in the figure to 0.1 mM

6) Paragraph 3.2. Arguing that it increases resistance (Rct) and states that it improves electron transfer when these data indicate otherwise.

Response: The Rct was increased a little but the performance of electrode surface was increased after modification, which was confirmed by the CV results (Figure 3B).

7) Section 3.2. Figure 3a and 3b I would use the same colors to indicate the curves.

Response: Thank you very much for your kind suggestion. We have changed the colors in the revised manuscript.

8) Figure 6 insert the corresponding letter inside the figures

Response: Thank you very much for your kind suggestion, we have inserted the corresponding letter inside the figures of Figure 6 in the revised manuscript.

9) Section 3.4.3 To better understand the sensor’s ability to detect an analyte in the presence of others, I would insert the relative standard deviation calculation (RSD%) which shows that there is no significant variation in the signals of fixed concentration analytes compared to the increase in the concentration of the test compound.

Response: Thank you very much for your kind suggestion. We have added RSD values in 3.4.3 in the revised manuscript.

10) Why was the interference study conducted with another electrochemical technique and not with the same (DPV) used for analysis.

Response: The amperometric detection is not suitable for measuring the four analytes simultaneously as the working potential should be more positive than all the oxidation peaks, but for the interference study, this technique was very convenient and clear.

11) There is no DPV analysis of all analytes (DA, UA, trp, TP) simultaneously increasing the concentration of all compounds constantly.

Response: We are very sorry for this negligence. As it is summer vacation now and the students are not in the lab, we cannot do the DPV analysis of all analytes simultaneously with the concentration in such a short time.

Round 2

Reviewer 1 Report

The consistent improvements in line with previous comments are considered to change my conclusion in this step with ”Accept in the present form”

Reviewer 2 Report

The authors answered the questions posed satisfactorily with the exception of the last question. However, this test could have served to enrich the manuscript, but given the simultaneous tests at fixed concentrations, this can be avoided. That said, the manuscript can be accepted in my opinion.